# Phenotype and molecular signature of CD8$^+$ T cell subsets in T cell- mediated rejections after kidney transplantation

Eun Jeong Ko[1,2ʘ], Jung-Woo Seo[3ʘ], Kyoung Woon Kim[1], Bo-Mi Kim[1], Jang-Hee Cho[4], Chan-Duck Kim[4], Junhee Seok[5], Chul Woo Yang[1,2], Sang-Ho Lee[6‡*], Byung Ha Chung[1,2‡*]

**1** Convergent Research Consortium for Immunologic Disease, Seoul St. Mary's Hospital, College of Medicine, The Catholic University of Korea, Seoul, Korea, **2** Department of Internal Medicine, College of Medicine, The Catholic University of Korea, Seoul, Korea, **3** Department of Core Research Laboratory, Medical Science Research Institute, Kyung Hee University Hospital at Gangdong, Seoul, Korea, **4** Department of Internal Medicine, Kyungpook National University School of Medicine, Daegu, Korea, **5** School of Electrical Engineering, Korea University, Seoul, South Korea, **6** Department of Internal Medicine, College of Medicine, Kyung Hee University, Seoul, Korea

ʘ These authors contributed equally to this work.
‡ These authors also contributed equally to this work.
* chungbh@catholic.ac.kr (CBH); lshkidney@khu.ac.kr (LSH)

**Data Availability Statement:** All relevant data are within the paper and its Supporting Information files.

## Abstract

We investigated the phenotype and molecular signatures of CD8$^+$ T cell subsets in kidney-transplant recipients (KTRs) with biopsy-proven T cell-mediated rejection (TCMR). We included 121 KTRs and divided them into three groups according to the pathologic or clinical diagnosis: Normal biopsy control (NC)($n = 32$), TCMR ($n = 50$), and long-term graft survival (LTGS)($n = 39$). We used flowcytometry and microarray to analyze the phenotype and molecular signatures of CD8$^+$ T cell subsets using peripheral blood from those patients and analyzed significant gene expressions according to CD8$^+$ T cell subsets. We investigated whether the analysis of CD8$^+$ T cell subsets is useful for predicting the development of TCMR. CCR7$^+$CD8$^+$ T cells significantly decreased, but CD28$^{null}$CD57$^+$CD8$^+$ T cells and CCR7$^-$CD45RA$^+$CD8$^+$ T cells showed an increase in the TCMR group compared to other groups ($p<0.05$ for each); hence CCR7$^+$CD8$^+$ T cells showed significant negative correlations to both effector CD8$^+$ T cells. We identified genes significantly associated with the change of CCR7$^+$CD8$^+$ T, CCR7$^-$CD45RA$^+$CD8$^+$ T, and CD28$^{null}$CD57$^+$CD8$^+$ T cells in an *ex vivo* study and found that most of them were included in the significant genes on *in vitro* CCR7$^+$CD8$^+$ T cells. Finally, the decrease of CCR7$^+$CD8$^+$ T cells relative to CD28$^{null}$CD57$^+$ T or CCR7$^-$CD45RA$^+$CD8$^+$ T cells can predict TCMR significantly in the whole clinical cohort. In conclusion, phenotype and molecular signature of CD8$^+$ T subsets showed a significant relationship to the development of TCMR; hence monitoring of CD8$^+$ T cell subsets may be a useful for predicting TCMR in KTRs.

**Funding:** BH Chung, CD Kim, and SH Lee received a grant from the Korean Health Technology R&D Project, Ministry of Health & Welfare, Republic of Korea (HI13C1232) and EJ Ko received a grant from the Bio & Medical Technology Development Program of the National Research Foundation (NRF) funded by the Ministry of Science & ICT (2018M3A9E802151). The funders had no role in study design, data collection and analysis, decision to publish, or preparation of the manuscript.

**Competing interests:** The authors have declared that no competing interests exist.

**Abbreviations:** AR, acute rejection; BPAR, biopsy-proven acute rejection; FDR, False discovery rate; KT, kidney transplantation; KTR, kidney-transplant recipient; LTGS, long-term graft survival; NC, normal biopsy control; PBMC, peripheral blood mononuclear cells; TCMR, T cell-mediated rejection.

# Introduction

**After kidney transplantation (KT), CD8⁺** T cells have an important role in the development of the allograft rejection process, not only by direct invasion to allograft tissue, but also by recruitment and activation of other types of immune cells. [1] Indeed, markers for the activation of CD8⁺ T cells can be detected in the peripheral blood isolated from kidney-transplant recipients (KTRs); especially, CD8⁺ T cell subsets belonging to the terminally-differentiated effector-cell state are known to be involved in the process of allograft rejection. [2–5] In contrast, CD8⁺ T cell subtypes that display a naïve cell state can be involved in an "anti-rejection" process by regulation of effector T cells. [6–8] Therefore, it is possible that the dynamics of CD8⁺ T subsets in the peripheral blood can show a significant change according to "rejection" and "stable" state; hence it has been proposed that monitoring of CD8⁺ T cells subsets may be useful for detecting acute allograft rejection. [3, 9, 10]

In our previous studies, we investigated the role of CD8⁺ T cell subsets, especially CCR7⁺CD8⁺ T cells, the naïve T cell, in regard to the suppression of kidney allograft rejection. [11, 12] We found that this cell type has a suppressive effect on effector T cell subsets in *in vitro* study. Also, its proportion in peripheral blood was decreased in kidney-transplant recipients (KTRs) with T cell-mediated rejection (TCMR) compared to the normal-biopsy control (NC) groups. In contrast, effector T cell types, such as CD28^null^CD57⁺CD8⁺ T cells (immune senescent T cells), CCR7⁻CD45RA⁺CD8⁺ T cells (TEMRA), which are known to be involved in allograft rejection, were significantly increased in patients with acute rejection [3–5]. These results suggest that the phenotype analysis of CD8⁺ T cell subsets, especially the relative proportions between CCR7⁺CD8⁺ T cells and other effector CD8⁺ T cells, may be associated with the development of acute allograft rejection. In addition, peripheral blood transcripts apparently can reflect the systemic immune status or several critical clinical conditions. [13–16]

Based on the above background, we intended to investigate the dynamics of CD8⁺ T cell subsets, including CCR7⁺CD8⁺ T cells along with CD28^null^CD57⁺CD8⁺ T and CCR7⁻CD45RA⁺CD8⁺ T cells, in KTRs with TCMR compared to those with normal biopsy (NC) or long-term stable allograft survival (LTGS). We also investigated the association between CD8⁺ T cell subsets and molecular signatures obtained by means of transcript analysis using a microarray in those patients and attempted to infer changes in peripheral- blood transcripts with the change of T cell subsets during acute allograft rejection after KT.

# Materials and methods

## Patients and clinical information

In an *ex vivo* study to compare CD8⁺ T cell subsets among clinical groups, peripheral-blood mononuclear-cell (PBMC) samples were chosen from the ARTKT-1 (assessment of immunologic risk and tolerance in kidney transplantation) study, a cross-sectional sample collection study of KTRs who had received kidney allograft biopsy or who had long-term allograft survival (LTGS) with stable allograft function (MDRD eGFR ≥ 50 mL/min/1.73 m²) over ten years at four different transplant centers (Kyoung Hee University Hospital at Gangdong, Kyung Hee University Hospital, Kyungpook National University Hospital, Seoul St. Mary's Hospital of Catholic University of Korea) from August 2013 to July 2015. [17–20] ARTKT-1 was used only to identify participants and access kidney tissue. Among the PBMC samples collected for the ARTKT-1 study, we used a total of 121 samples from 32 patients with normal biopsy without any evidence of rejection (NC group) and 50 patients who showed T cell-mediated rejection (TCMR) on allograft biopsy with Banff classification assessed by a single pathologist (TCMR group) [21] and 39 patients with LTGS for this study. We did not include

**Table 1. Baseline characteristics of the kidney transplant recipients included in ex vivo study.**

| | NC (n = 32) | TCMR (n = 50) | LTGS (n = 39) | P |
|---|---|---|---|---|
| Age (year) | 41.5 ± 14.4 | 48.9 ± 11.6 | 56.0 ± 8.7 | <0.001 |
| Male, n (%) | 24 (75) | 31 (62) | 18 (46) | 0.045 |
| Post-transplant month | 6.6 ± 1.4 | 18.0 ± 20.0 | 204.5 ± 84.8 | <0.001 |
| MDRD eGFR (mL/min/1.73 m$^2$) | 69.6 ± 37.8 | 32.5 ± 15.4 | 69.5 ± 16.5 | <0.001 |
| HLA mismatch number | 3.9 ± 1.5 | 3.2 ± 1.7 | 2.2 ± 1.3 | <0.001 |
| ABO incompatible KT, n (%) | 8 (25) | 13 (26) | 0 (0) | <0.001 |
| Previous TCMR, n (%) | 1 (3) | 24 (48) | 0 (0) | <0.001 |
| Pretransplant DSA, n (%) | 5 (16) | 10 (20) | 1 (3) | <0.001 |
| Re-transplant, n (%) | 5 (16) | 5 (10) | 1 (3) | 0.156 |
| Indication for biopsy | | | | |
| Protocol biopsy, n (%) | 3 (9) | 7 (14) | N/A | <0.001 |
| Indicated biopsy, n (%) | 29 (91) | 43 (86) | N/A | |
| Induction IS | | | | |
| Basiliximab, n (%) | 30 (94) | 43 (86) | 34 (87) | 0.540 |
| Anti-thymocyte globulin, n (%) | 2 (6) | 7 (14) | 5 (13) | |
| Maintenance IS | | | | |
| Tacrolimus, n (%) | 31 (97) | 37 (74) | 10 (26) | <0.001 |
| Cyclosporin, n (%) | 0 (0) | 10 (20) | 18 (46) | <0.001 |
| mTOR inhibitor, n (%) | 2 (6) | 2 (4) | 3 (8) | 0.754 |
| Mycophenolate Mofetil, n (%) | 29 (91) | 40 (80) | 10 (26) | <0.001 |
| Steroid, n (%) | 31 (97) | 42 (84) | 20 (51) | <0.001 |
| Azathioprine, n (%) | 0 (0) | 0 (0) | 4 (10.3) | 0.013 |
| Donor information | | | | |
| Deceased donor, n (%) | 7 (22) | 19 (38) | 6 (15) | 0.014 |
| Donor age | 48.6 ± 7.8 | 47.1 ± 12.2 | 35.5 ± 11.9 | <0.001 |
| Donor gender (male, n (%)) | 15 (47) | 29 (58) | 21 (54) | 0.615 |

DSA, donor specific antibody; eGFR, estimated glomerular filtration rate; IS, immune suppression; KT, kidney transplantation; LTGS, long term graft survival; MDRD, Modification of diet in renal disease; NC, Normal biopsy control; TCMR, T cell mediated rejection

patients who took any other solid organ transplantation in this study. The baseline characteristics of both groups are presented in Table 1. All participants provided written informed consent in accordance with the Declaration of Helsinki. The study protocol was registered in the Clinical Research Information Service (CRIS Registration Number: KCT0001010), and was approved by the Institutional Review Board of each participating hospital. [Seoul St. Mary's Hospital (KC13TNMI0701); Kyungpook National University Hospital (2013-10-010); Kyung Hee Neo Medical Center (IRB No. 2012-01-030)].

## Flowcytometric analysis of peripheral-blood CD8$^+$ T cells isolated from kidney-transplant recipients

From 121 KTRs, PBMCs (1 x 10$^6$ cells/mL) were prepared from heparinized blood by Ficoll–Hypaque (GE Healthcare) density-gradient centrifugation. Cells were stored frozen at each center within 1 hour after the sampling of peripheral blood. They transported to our center for flowcytometric analysis. Cells were cultured as described previously [12, 22]. In brief, a cell suspension of 1 x 10$^6$ cells/mL was prepared in RPMI1640 medium supplemented with 10% FCS, 100 U/mL penicillin, 100 mg/mL streptomycin, and 2 mM L-glutamine. The cells were

surface-stained with different combinations of the following monoclonal antibodies: CD8–APC (SK1, IgG1,κ; BD), CCR7-strepavidin (3D12, IgG2a, κ), CD45RA–FITC (HI100, IgG2b, k; BD), CD28-PE (CD28.2, IgG1,κ, eBioscience) and CD57-FITC (TB01, IgM, eBioscience). Appropriate isotype controls were used for gating purposes. Cells were analyzed using a FACS Calibur flow cytometer (BD Biosciences). We analyzed the data using FlowJo software (Tree Star).

## Relationships between transcriptome expression and T cell subsets

Previously, we did both microarray and flowcytometry analysis in 153 KTRs belonging to ARTKT-1 study. [18] For this study, we used microarray and flowcytometry data of 108 KTRs in whom the data for CCR7+CD8+ T, CCR7-CD45RA+CD8+ T, and CD28$^{null}$CD57+CD8+ T cells were available. The microarray analysis using RNA isolated from peripheral blood from KTRs was described previously. [18] Briefly, peripheral blood was collected in 2.5 mL PAXgene$^{TM}$ Blood RNA Tubes (PreAnalytiX, Qiagen) and total RNA was extracted from PAXgene samples using Paxgene Blood miRNA Kit (PreAnalytiX, Qiagen) according to manufacturer's protocol. We measured quantity and quality of total RNA using Agilent's 2100 Bioanalyzer. We used the universal human reference RNA (Agilent Technology, USA) as control for two-color microarray-based gene-expression analysis and synthesized the target cRNA probes and hybridization using Agilent's Low RNA Input Linear Amplification kit (Agilent Technology, USA) according to the manufacturer's instructions. The hybridized images were scanned using Agilent's DNA microarray scanner and quantified with Feature Extraction Software (Agilent Technology, Palo Alto, CA). All data normalization and selection of fold-changed genes were done using GeneSpringGX 7.3 (Agilent Technology, USA). The averages of normalized ratios were calculated by dividing the average of normalized signal channel intensity by the average of normalized control channel intensity.

We identified significant changes in gene expression associated with cell types using the SAM (Significance Analysis of Microarray) R package [23]. Briefly, expression data in 101 samples matched to cell-phenotype data were normalized by rescaling mean 0 and standard deviation 1, and then SAM analysis was used for cell-type expressivity as a quantitative response variable. The false discovery rates were obtained from 1,000 permutations. We discovered significantly differently expressed genes (FDR < 0.05) for each of the cell types.

## Microarray analysis using isolated CCR7+ CD8+ T or CCR7- CD8+ T cells from healthy volunteers

**Isolation of CCR7+ CD8+ T or CCR7- CD8+ T cells and extraction of RNA.** From three healthy volunteers, PBMCs (1 x 10$^6$ cells/mL) were prepared from heparinized blood (10cc) by Ficoll–Hypaque (GE Healthcare) density-gradient centrifugation. To expand CCR7+CD8+ T cells, isolated PBMCs were stimulated using anti-CD3, IL-15, IL-2, and retinoic acid. Cells were cultured as described previously. [11] We pooled cells for microarray as opposed to single cell RNA sequencing. CCR7+ CD8+ T cells were purified by CD8–APC (SK1, IgG1,κ; BD) and CCR7-strepavidin (3D12, IgG2a, κ). The cells were sorted using an FACS Aria device (Becton Dickinson) or a MoFlo cell sorter (Beckman Coulter) to isolate CCR7+ CD8+ and CCR7- CD8+ T cells. We extracted mRNA from CCR7+CD8+ and CCR7- CD8+ T cells using the ReliaPrep™ RNA Miniprep Systems (Promega Corporation, Madison, WI, USA), according to the manufacturer's instructions. RNA purity and integrity were evaluated by ND-1000 Spectrophotometer (NanoDrop, Wilmington, DE, USA).

**Affymetrix whole transcript expression array method.** We carried out the Affymetrix Whole Transcript Expression array process according to the manufacturer's protocol (GeneChip WT Pico Reagent Kit). We synthesized cDNA using the GeneChip WT Pico

Amplification kit as described by the manufacturer. The sense cDNA was fragmented and bio-tin-labeled with TdT (terminal deoxynucleotidyl transferase) using the GeneChip WT Termi-nal labeling kit. Approximately 5.5 μg of labeled DNA target was hybridized to the Affymetrix GeneChip Human 2.0 ST Array at 45˚C for 16 hours. Hybridized arrays were washed and stained on a GeneChip Fluidics Station 450 and scanned on a GCS3000 Scanner (Affymetrix). Signal values were computed using the Affymetrix® GeneChip™ Command Console software.

### Raw data preparation and statistical analysis

Statistical analysis was done by using SPSS software (version 16.0; SPSS, Inc., Chicago, IL, USA). Values between groups were compared using one-way analysis of variance. For categorical vari-ables, chi-square frequency analysis was used. The results are presented as mean ± standard deviation (SD). $P$ values $< 0.05$ were considered significant. For microarray analysis, raw data were extracted automatically in the Affymetrix data-extraction protocol using software provided by Affymetrix GeneChip® Command Console® Software (AGCC). After we imported CEL files, we summarized and normalized the data with the robust multi-average (RMA) method implemented in Affymetrix® Expression Console™ Software (EC). We exported the results with gene-level RMA analysis and carried out differently expressed gene (DEG) analysis. Statistical significance of the expression data was assessed using fold change and an LPE test in which the null hypothesis was that no difference exists between groups. The false discovery rate (FDR) was controlled by adjusting the $p$ value using the Benjamini-Hochberg algorithm. For a DEG set, we did hierarchical cluster analysis using complete linkage and Euclidean distance as a measure of similarity. All data analysis and visualization of differently expressed genes was done using R 3.1.2 (www.r-project.org). For the analysis of the relationships between transcriptome expres-sion and T cell subsets, a score for each gene of a statistically significant change in gene expres-sion relative to cell type was established by $t$ tests, and the "$q$ value" for each gene was the lowest false discovery rate (FDR). Significant genes were selected by high score and $q < 0.05$.

## Results

### Ex vivo analysis of CD8$^+$ T cell subset in KTRs

Fig 1A shows representative flow cytometric data for lymphocytes, CD8$^+$ T, CCR7$^+$CD45RA$^+$ CD8$^+$ T, CCR7$^+$CD45RA$^-$CD8$^+$ T, CCR7$^+$CD8$^+$T, CCR7$^-$CD45RA$^-$CD8$^+$ T, CCR7$^-$CD45RA$^+$ CD8$^+$T and CD28$^{null}$CD57$^+$CD8$^+$T cells in KTRs. The percentage of lymphocytes was signifi-cantly decreased in the TCMR group in comparison with the NC group (p $< 0.01$) and LTGS group ($p < 0.05$) (Fig 1B). In contrast, the percentage of CD8$^+$ out of lymphocytes was signifi-cantly increased in the TCMR group in comparison with the NC group or LTGS group ($p < 0.01$ for each) (Fig 1C). The percentage of CCR7$^+$CD45RA$^+$CD8$^+$ T cells and CCR7$^+$CD45RA$^-$ CD8$^+$ T cells was significantly decreased in the TCMR group in comparison with the NC group ($p < 0.01$) (Fig 1D and 1E). Therefore, the percentage of CCR7$^+$CD8$^+$ T cells was also significantly decreased in the TCMR group in comparison with the NC group (p $< 0.01$) and LTGS group ($p < 0.001$) (Fig 1F). In contrast, the proportion of CCR7$^-$CD45RA$^+$CD8$^+$ T and CD28$^{null}$CD57$^+$CD8$^+$ T cells were significantly higher in the TCMR group than in the NC group ($p < 0.05$) or LTGS group, respectively ($p < 0.05$ for all) (Fig 1H and 1I).

### Association between CCR7$^+$CD8$^+$ T cells and other effector T cells in KTRs and in vitro condition

In the *ex vivo* study in KTRs, the proportion of CCR7$^+$CD8$^+$ T cells showed a significant nega-tive correlation with CD28$^{null}$CD57$^+$CD8$^+$ T ($p < 0.001$, $R^2 = 0.38$), and CCR7$^-$CD45RA$^+$CD8$^+$

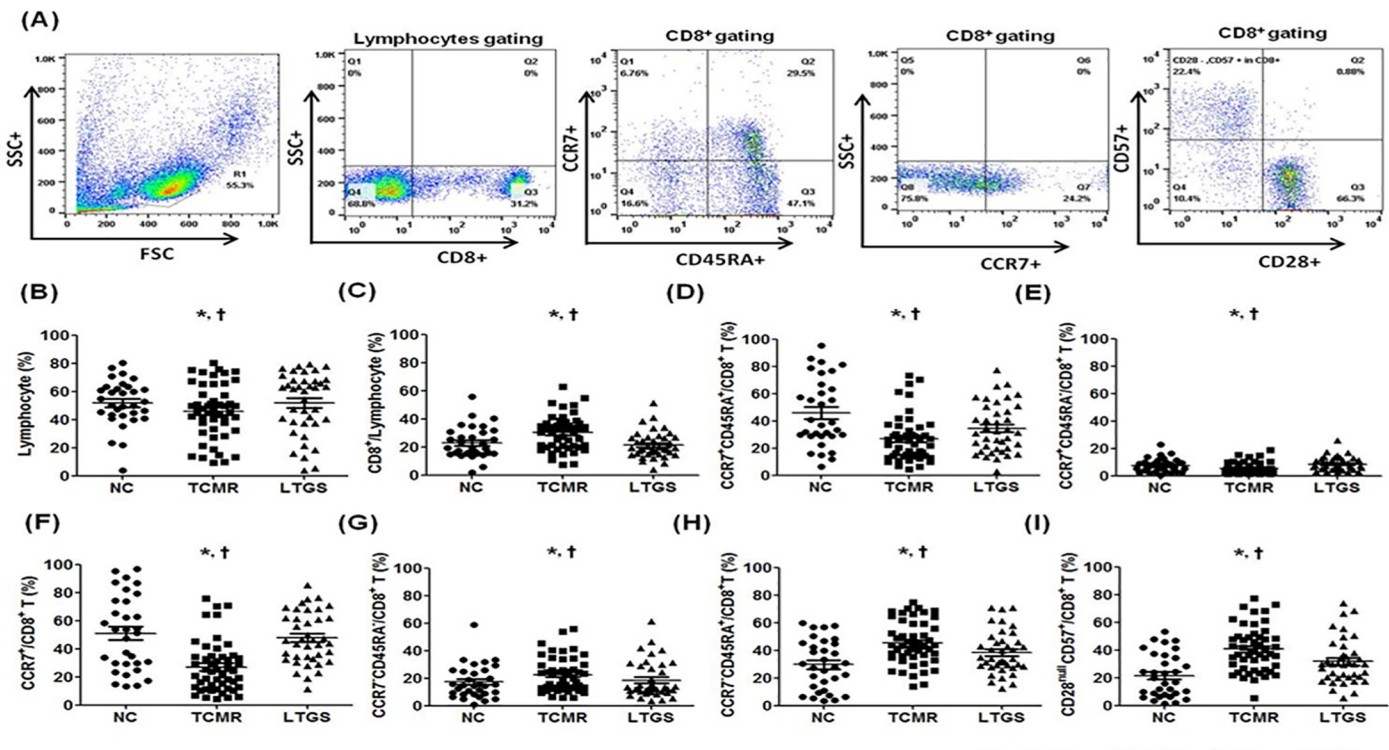

**Fig 1. Comparison of CD8+ T cell subset among NC, TCMR and LTGS groups.** (A) PBMCs were stained with anti-CD8–APC, anti-CCR7 strepavidin, anti-CD45RA FITC, anti-CD28-PE and anti-CD57-FITC antibodies. CD8+ T cells were gated for further analysis. (B-I) Proportion (%) of (B) Lymphocytes, (C) CD8+ T cells/ lymphocytes, (D) CCR7+CD45RA+CD8+ T cells, (E) CCR7+CD45RA-CD8+ T cells, (F) CCR7+CD8+ T cells, (G) CCR7-CD45RA-CD8+ T cells, (H) CCR7-CD45RA+CD8+ T cells, (I) CD28null CD57+CD8+ T cells in each patient group. *$p < 0.05$ vs. NC, †$p < 0.05$ vs. LTGS. Abbreviations; LTGS, long-term graft survival; NC, Normal control; TCMR, T cell mediated rejection.

($p < 0.001$, $R^2 = 0.51$) (Fig 2A and 2B). Therefore, when we compared the log transformation value of the ratio between two types of effector T cells (CD28null CD57+ CD8+ T or CCR7−CD45RA+CD8+ T) and CCR7+CD8+ T cells, it was significantly lower in the TCMR group than in the NC or LTGS group ($p < 0.05$ for both)(Fig 2C and 2D)

In an *in vitro* study on CCR7+CD8+ T cells induction protocol using PBMCs isolated from healthy volunteers, the proportion of CCR7+CD8+ T cells was significantly higher on the CCR7+CD8+ induction protocol than Nil ($p < 0.001$) (Fig 2C). In contrast, the CCR7+CD8+ T cells induction protocol significantly reduced the proportion of CD28null CD57+CD8+ ($p < 0.01$) and CCR7−CD45RA+CD8+ in contrast with Nil (Fig 2E–2G).

## Association analysis between peripheral transcriptome and ex vivo CD8+ T cell subsets

We investigated significant changes of the gene expression in peripheral blood associated with CCR7+CD8+ T, CCR7-CD45RA+CD8+ T, and CD28null CD57+CD8+ T cells in an *ex vivo* study. We identified 13 up-regulated genes but no down-regulated genes in KTRs with the high proportion of CCR7+CD8+ T cells (S1 Table). We also identified eight increased and three decreased genes in KTRs along with the change of proportion of CCR7-CD45RA+CD8+ T and found 124 up-regulated and 19 down-regulated genes in those along with the change of CD28null CD57+CD8+ T cells (S2 and S3 Tables).

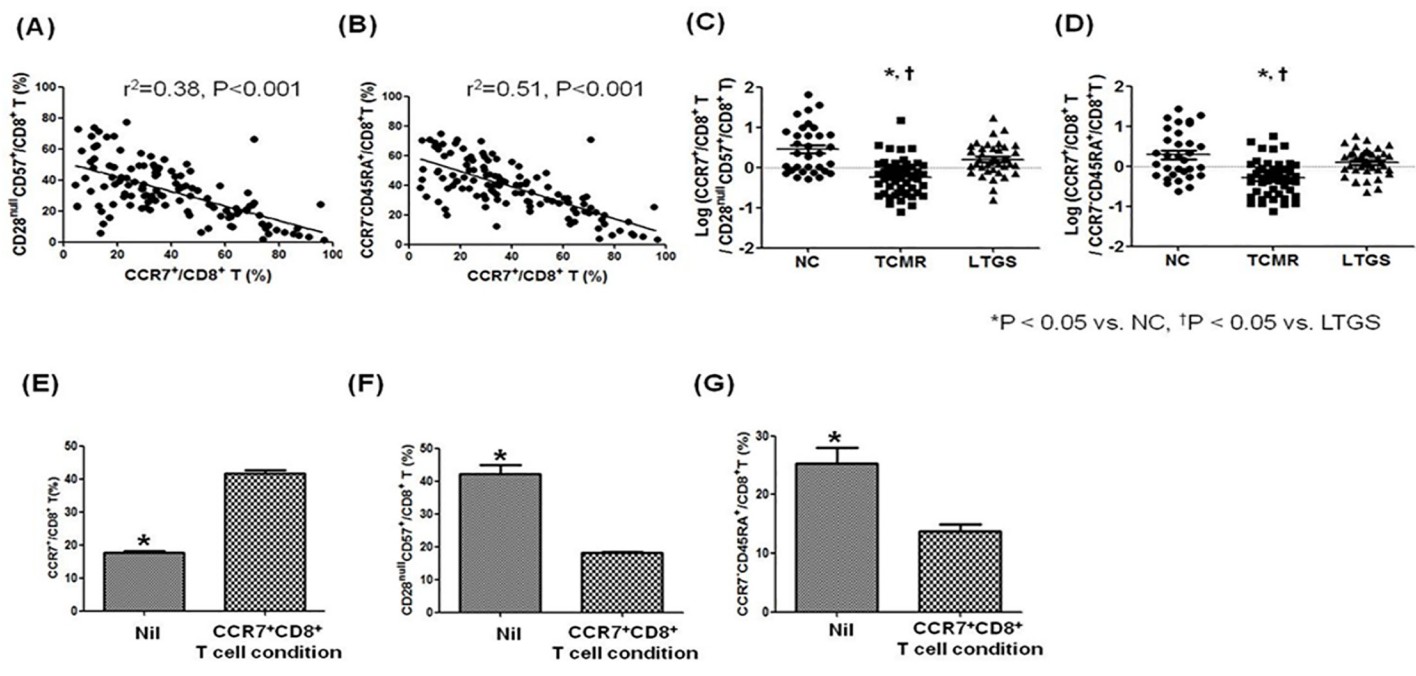

**Fig 2. Association between CCR7+/CD8+ T cells and effector T cell subsets in an *in vitro* and in an *ex vivo* study.** PBMCs were stained with anti-CD8–APC, anti-CCR7 strepavidin, anti-CD45RA FITC, anti-CD28-PE and anti-CD57-FITC antibodies. Lymphocytes were gated for further analysis. **(A)** The proportion (%) of CCR7+CD8+ T cells showed a significant negative correlation with the proportion (%) of CD28null CD57+CD8+ T cells ($p < 0.001$, $r^2 = 0.38$). **(B)** The proportion (%) of CCR7+CD8+ T cells showed a significant negative correlation with the proportion (%) of CCR7-CD45RA+CD8+ T cells ($p < 0.001$, $r^2 = 0.51$). **(C)** Comparison of log (CCR7+CD8+ T cells / CD28nullCD57+CD8+ T cells) in each patient group. **(D)** Comparison of log (CCR7+CD8+ T / CCR7-CD45RA+CD8+T cells) in each patient group * $p < 0.05$ vs. NC, † $p < 0.05$ vs. LTGS. In an *in vitro* study on CCR7+CD8+ T cells induction protocol, PBMCs were collected from healthy individuals, plated at $2 \times 10^5$ cells per well, and stimulated with anti-CD3 Abs (0.1 μg/ml), recombinant IL-15 (20 ng/ml), IL-2 (20 ng/ml), and retinoic acid (1 μg/ml). On day 3, cells were harvested, stained with antibodies specific to CD8, CCR7, CD45RA, CD28 and CD57, and analyzed by flow cytometry. **(E)** Proportion (%) of CCR7+CD8+ T cells, (F) Proportion (%) of CD28nullCD57+CD8+ T cells, **(G)** Proportion (%) of CCR7-CD45RA+CD8+ T cells on CCR7+CD8+ T cells induction protocol. Bars represent the median with range. * $p < 0.05$ vs. CCR7+CD8+ T cell condition. Abbreviations; LTGS, long-term graft survival; NC, Normal control; TCMR, T cell mediated rejection.

## Microarray analysis of RNA from isolated CCR7+CD8+ T and CCR7-CD8+ T cells in healthy volunteers

We did microarray analysis on CCR7+CD8+ T cells compared with CCR7-CD8+ T isolated and induced using the PBMC from the same donors (*n* = 3) (Fig 3A) and identified 992 differently expressed genes. Moreover, comparison of the leading-edge gene set (the core set of genes that account for this enrichment) from each T cell population distinguished a core of 450 up-regulated (see S4 Table) and 542 down-regulated (see S5 Table) genes that were commonly enriched in all CCR7+-expressing CD8+ T cells and therefore define the CCR7+CD8+-associated transcriptional signature. Genes whose expression levels were higher than the assumed threshold (up-regulated > 1.5-fold and down-regulated < 1.5-fold) were visualized using the scatter plot method (Fig 3B).

In addition, we investigated the relationships between the genes expressed along CD8+ T cell subsets on *ex vivo* and 992 genes expressed on *in vitro* CCR7+CD8+ T cells. Out of the 13 increasingly changed genes along with the *ex vivo* CCR7+CD8+ T cells, six genes were included in the up-regulated genes on *in vitro* CCR7+CD8+ T cells (Table 2). Out of the eight up-regulated genes along with the *ex vivo* CCR7-CD45+CD8+ T cells (S2 Table), five genes were included in the down-expressed genes on *in vitro* CCR7+CD8+ T cells (Table 3). In addition,

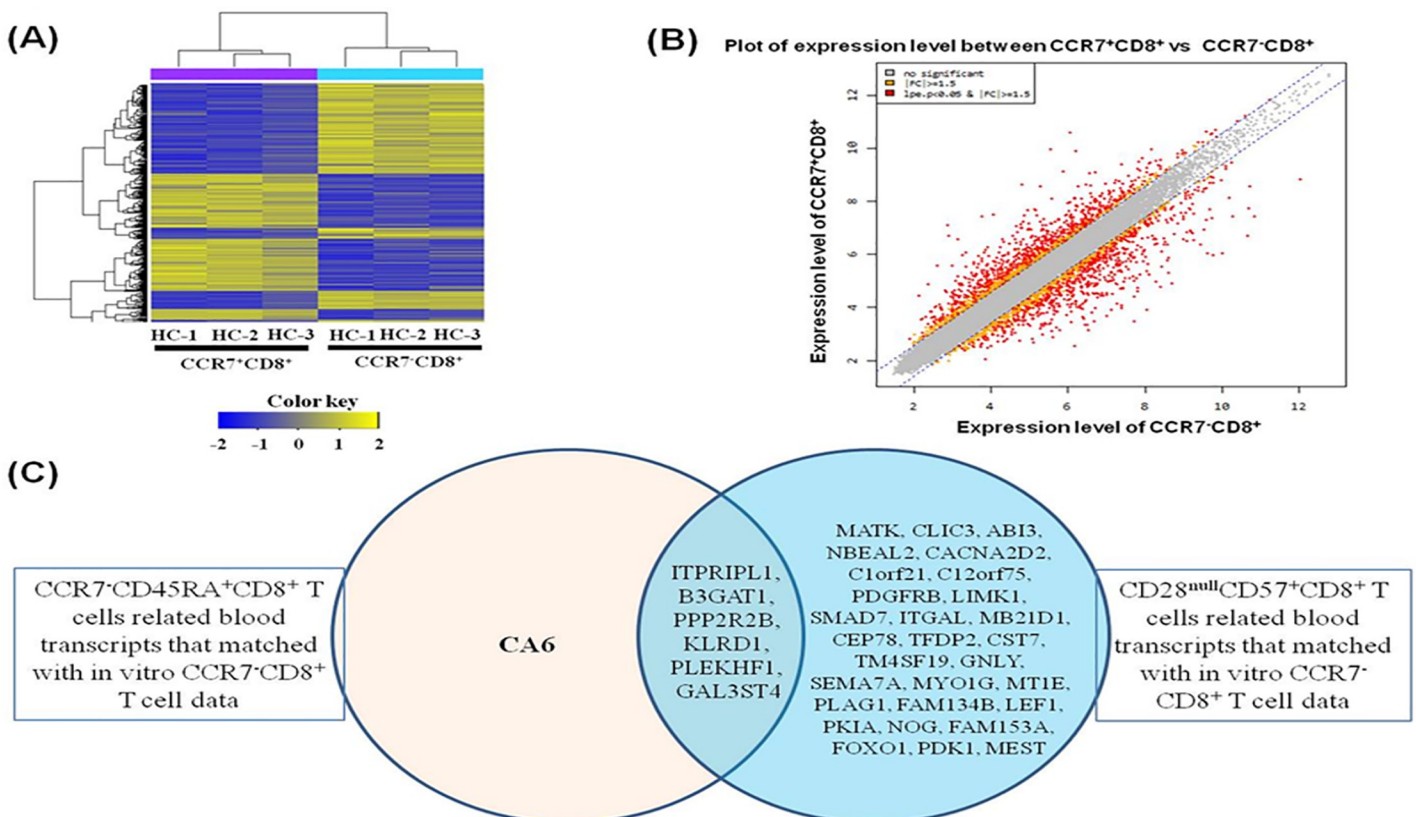

**Fig 3. Gene expression in CCR7+CD8+ T cells and CCR7-CD8+ T cells using microarray.** (A) Hierarchical clustering of gene expression in CCR7+CD8+ T cells and CCR7-CD8+ T cells. Heatmap is showing 992 significantly ($p < 0.05$) differently expressed transcripts between CCR7+CD8+ T cells and CCR7-CD8+ T cells in three donors. The 992 genes were selected for this analysis by the criteria described in Materials and Methods. Expression levels are normalized for each gene and shown by color, with yellow representing high expression and blue representing low expression. (B) Scatter plot of expression level between CCR7+CD8+ T cells and CCR7-CD8+ T cells. (C) The overlap between the genes expressed on *ex vivo* CCR7-CD45+CD8+ T cells matched with in vitro CCR7-CD8+ T cell data and CD28nullCD57+CD8+ T cells related blood transcripts that matched with *in vitro* CCR7-CD8+ T cell data.

out of three down-regulated genes along with the *ex vivo* CCR7-CD45+CD8+ T cells (S2 Table), two genes were included in the up-expressed genes on *in vitro* CCR7+CD8+ T cells (Table 3). Also, 25 out of 124 up-regulated genes and 10 out of 19 down-regulated genes along with *ex vivo* CD28nullCD57+CD8+ T cells (S3 Table) were included in the down-regulated or up-regulated genes on *in vitro* CCR7+CD8+ T cells, respectively, as shown in Table 4. The genes expressed on *ex vivo* CCR7-CD45+CD8+ T cells were correlated with those on *ex vivo* CD28nullCD57+CD8+ T cells, except for CA6 (Fig 3C). These results showed that CCR7+CD8+ T cells are negatively correlated with CCR7-CD45RA+CD8+ T and CD28nullCD57+CD8+ T cells in both cell phenotype and molecular signature.

**Table 2. CCR7+CD8+ T cells related blood transcripts that matched with in vitro CCR7+CD8+ T cell data.**

| Gene Symbol | Score (d) | q-value (%) |
| --- | --- | --- |
| CA6 | 3.957 | 0 |
| EDAR | 3.674 | 0 |
| NOG | 3.585 | 0 |
| GCNT4 | 3.442 | 0 |
| LEF1 | 3.380 | 0 |
| LRRN3 | 3.147 | 4.136 |

**Table 3. CCR7$^-$CD45RA$^+$CD8$^+$ T cells related blood transcripts that matched with in vitro CCR7$^+$CD8$^+$ T cell data.**

| Gene Symbol | Score (d) | q-value (%) |
|---|---|---|
| ITPRIPL1 | 4.373 | 0 |
| B3GAT1 | 4.258 | 0 |
| PPP2R2B | 4.058 | 0 |
| KLRD1 | 3.826 | 0 |
| PLEKHF1 | 3.748 | 0 |
| CA6 | -3.654 | 0 |
| GAL3ST4 | -3.552 | 0 |

## Receiver Operating Characteristic (ROC) curve analysis to evaluate the ability of the ratios between CD8 T cell subsets to predict AR

We evaluated the diagnostic power of CCR7$^+$CD8$^+$T, CCR7$^-$CD45RA$^+$CD8$^+$T, CD28$^{null}$CD57$^+$CD8$^+$ T cells, the ratio between CCR7$^+$CD8$^+$T and CCR7$^-$CD45RA$^+$CD8$^+$T and between CCR7$^+$CD8$^+$T and CD28$^{null}$CD57$^+$CD8$^+$ T cells to distinguish the acute rejection state from normal biopsy or long-term stable condition using the AUC, which was found via ROC curve analysis (Fig 4). The AUCs of CCR7$^+$CD8$^+$T, CD28$^{null}$CD57$^+$CD8$^+$ T cells, and CCR7$^-$CD45RA$^+$CD8$^+$T were 0.785, 0.728, and 0.675 respectively. The AUC value was increased to 0.768 and 0.800 when we used the ratio between CCR7$^+$CD8$^+$T and CCR7$^-$CD45RA$^+$CD8$^+$T or between CCR7$^+$CD8$^+$T and CD28$^{null}$CD57$^+$CD8$^+$ T cells respectively. After the integration of both ratios, they did not increase from the AUCs of the ratio between CCR7$^+$CD8$^+$T and CD28$^{null}$CD57$^+$CD8$^+$ T cells.

## Discussion

In this study, we analyzed various CD8$^+$ T cell subsets using flow cytometry and a microarray method to see the relationship between regulatory and effector CD8$^+$ T cell subsets in kidney transplant recipients with acute rejection. Finally, we found that CCR7$^+$CD8$^+$ T showed a negative relationship to inflammatory subsets (CD28$^{null}$CD57$^+$CD8$^+$ T and CCR7$^-$CD45RA$^+$CD8$^+$ T cells), not only in the cell-proportion results by flow cytometry but also in the transcriptomic expression by microarray. Therefore, our results showed that combined analysis of the CD8$^+$ T cell subset can be a useful tool for detecting the development of acute rejection.

First, we tried to compare the proportion of each CD8+ T cell subset in the different clinical groups, NC, AR, and LTGS. Previously, we found that CD8$^+$CCR7$^+$ T cells showed an immune-regulatory function on the other effector T cells; that function showed a negative relationship to effector CD8+ T cells involved in the development of acute rejection in a few patients [11]. In this study, we used a larger patient group and found the decrease of CD8$^+$CCR7$^+$ T cells and increase of effector CD8$^+$ T cells, such as CD28$^{null}$CD57$^+$CD8$^+$ T and CCR7$^-$CD45RA$^+$CD8$^+$ T cells in KTRs with AR in comparison with those in the NC or LTGS groups in the *ex vivo* flowcytometric analysis using PBMCs isolated from KTRs. In addition, we found the negative relationship between CCR7$^+$CD8$^+$ T cells and other effector T cells, as our previous study [11]. We tried to confirm those relationships in an *in vitro* study. In this experiment, we used a previously established protocol, including anti-CD3, IL-15, IL-2, and retinoic acid, for the induction of CCR7$^+$CD8$^+$ T cells [24, 25] and found that with the increase of CCR7$^+$CD8$^+$ T cells, the proportion of CD28$^{null}$CD57$^+$CD8$^+$ T and CCR7$^-$CD45RA$^+$CD8$^+$

**Table 4. CD28^null^CD57+CD8+ T cells related blood transcripts that matched with in vitro CCR7+CD8+ T cell data.**

| Gene Symbol | Score (d) | q-value (%) |
|---|---|---|
| ITPRIPL1 | 4.672 | 0 |
| MATK | 4.320 | 0 |
| B3GAT1 | 4.216 | 0 |
| CLIC3 | 4.075 | 0 |
| ABI3 | 3.660 | 0 |
| NBEAL2 | 3.647 | 0 |
| CACNA2D2 | 3.480 | 2.46 |
| PPP2R2B | 3.476 | 2.46 |
| C1orf21 | 3.439 | 2.46 |
| KLRD1 | 3.419 | 2.46 |
| C12orf75 | 3.332 | 2.46 |
| PDGFRB | 3.307 | 2.46 |
| LIMK1 | 3.295 | 2.46 |
| SMAD7 | 3.273 | 2.46 |
| ITGAL | 3.270 | 2.46 |
| MB21D1 | 3.167 | 2.46 |
| CEP78 | 3.133 | 2.46 |
| TFDP2 | 3.108 | 2.46 |
| CST7 | 3.106 | 2.46 |
| TM4SF19 | 3.088 | 2.46 |
| GNLY | 3.055 | 2.46 |
| PLEKHF1 | 3.036 | 2.46 |
| SEMA7A | 3.028 | 2.46 |
| MYO1G | 3.004 | 2.46 |
| MT1E | 2.983 | 2.46 |
| PLAG1 | -4.151 | 0 |
| GAL3ST4 | -4.070 | 0 |
| FAM134B | -3.679 | 2.46 |
| LEF1 | -3.543 | 2.46 |
| PKIA | -3.420 | 2.46 |
| NOG | -3.418 | 2.46 |
| FAM153A | -3.412 | 2.46 |
| FOXO1 | -3.403 | 2.46 |
| PDK1 | -3.341 | 2.46 |
| MEST | -3.324 | 2.46 |

T cells showed a decrease in CCR7+CD8+ T cell induction condition in comparison with Nil condition.

Next, we did microarray analysis to profile genes in peripheral blood in KTRs using Agilent's Human Oligo Microarray 60K (V2) and identified significant genes changed along with the cell proportion of CD8+ T cell subsets by the *ex vivo* flowcytometric analysis. The changes in T cell subsets developed associated with acute rejection in KTRs have been frequently investigated, including in our own works [5, 22, 26]. In contrast, there has been little research on the molecular signatures representing the changes in T cell subsets. Recently, the new development of an assay for an RNA transcript enabled the analysis of the molecular signature of the immune cells specific to the disease status in KTRs [27–29]. In this study, candidate molecular

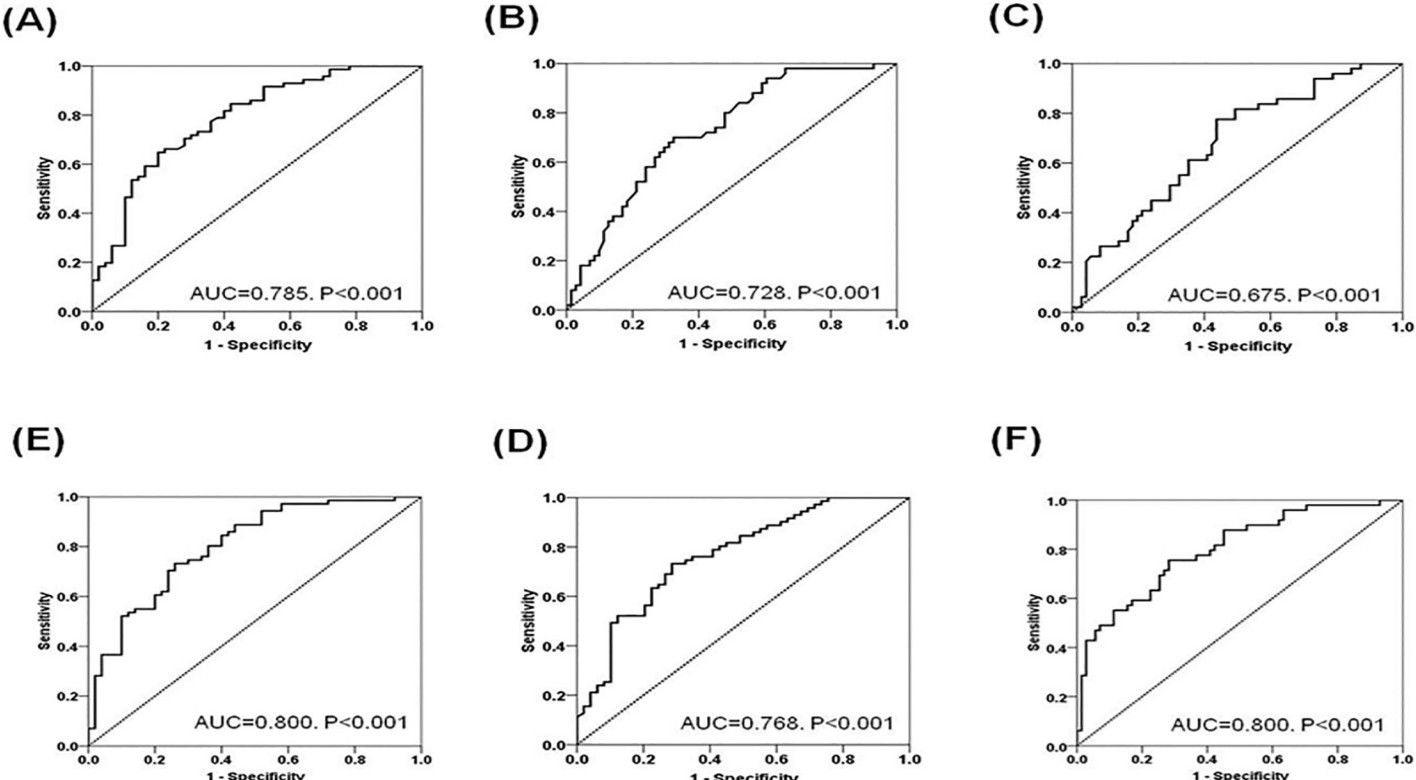

**Fig 4. Receiver operating characteristics curves to evaluate the discriminative power of the combination of CD8+ T cell subsets in distinguishing TCMR from the NC or LTGS groups. (A)** CCR7+CD8+ T **(B)** CD57+CD28nullCD8+ T **(C)** CD45RA+CCR7-CD8+ T **(D)** The ratio between CCR7+CD8+ T vs CD28nullCD57+CD8+ T **(E)** The ratio between CCR7+CD8+ T vs. CD45RA+CCR7-CD8+ T **(F)** Combination of the ratio between CCR7+CD8+ T vs CD28nullCD57+CD8+ T and the ratio between CCR7+CD8+ T vs CD45RA+CCR7-CD8+ T. Abbreviations; LTGS, long-term graft survival; NC, Normal control; TCMR, T cell mediated rejection.

signatures of rejection-specific T cell subsets were identified in peripheral-blood microarray analysis, and these molecular signatures were verified by *in vitro* analysis.

Interestingly, a significant portion of genes identified as related to the change of CCR7+CD8+ T, CCR7-CD45+CD8+ T, and CD28nullCD57+CD8+ T cells *ex vivo* were matched with CCR7+CD8+ T cells in *in vitro* transcript analysis (6/13 genes in CCR7+CD8+ T, 7/11 in CCR7-CD45+CD8+ T, and 35/143 in CD28nullCD57+CD8+ T cells) Moreover, the negative correlation between the CCR7+CD8+ T and effector T cell subset and also a positive correlation between CCR7-CD45+CD8+ T and CD28nullCD57+CD8+ T observed in phenotype analysis by flow cytometry was also maintained in the expression of these transcripts (Figs 2 and 3 and S1 Fig). All of the above findings suggest the reciprocal relationship between CCR7+CD8+ T cells and effector T cells subsets; hence they may suggest that the ratio between CCR7+CD8+ T cells and effector T cells subset can be significantly associated with development of TCMR rather than single-cell analysis.

Therefore, we appraised the significance of the ratio between CCR7+CD8+ T cells and other effector CD8+ T cells as well as single-cell results for the prediction of acute rejection. We found that the ratio between cell types showed better prediction for AR than did single-cell analysis, and the ratio between CCR7+CD8+ T and CD28nullCD57+CD8+ T showed the highest value for AR prediction. Unfortunately, the integration of the two ratio markers did not increase the predictive value over that of the ratio between CCR7+CD8+ T and CD28nullCD57+ CD8+ T itself. Indeed, CCR7-CD45RA+CD8+ T and CD28nullCD57+CD8+ T showed a

significant correlation with each other; hence many of them can overlap (S1 Fig). That's why integration of the two ratios did not increase the predictive value.

Our study has some limitations. We analyzed samples taken from a cross-sectional cohort; hence we did not investigate the dynamic changing pattern of each cell type. It will be necessary to observe the change of each cell type in a prospective cohort. Second, it was not validated in another cohort. Third, because of the inherent limitations of the entire transcriptome assay, too few transcripts for each rejection-specific T cell subset were identified. If it can be supplemented by means of a single-cell assay in the future, development of rejection-specific transcriptomic markers using peripheral blood will become possible. [30–32] In addition, three groups showed significant heterogeneity in terms of clinical characteristics. For example, patient age, type of immune suppressant, different post-transplant duration can impact on the result. Especially, induction immunosuppression, high-dose initial maintenance immunosuppression, and ABO desensitization may have contributed to observed findings in patients with less than 6 months from KT. Lastly, we did not show donor-specificity of T cell responses, which can limit the novelty of this study.

In conclusion, we found the relative decrease of CD8⁺ T cells with a regulatory function compared to effector CD8⁺ T cell subsets was the important phenomenon that can be detected in TCMR in comparison with NC or LTGS. We demonstrated this phenomenon in the phenotype analysis using flow cytometry, and also found that the distribution of CD8⁺ T cell subsets are correlated with molecular signatures by microarray transcript analysis. These findings suggest that combined monitoring of regulatory and effector CD8⁺ T cells subsets can be used as a surrogate marker of TCMR in KTRs.

## Supporting information

**S1 Fig. Association between CCR7-CD45RA⁺CD8⁺ T cells and CD28nullCD57⁺CD8⁺ T cells.** The proportion (%) of CCR7⁻CD45RA⁺CD8⁺ T cells showed a significant correlation with the proportion (%) of CD28$^{null}$CD57⁺CD8⁺ T cells ($p < 0.001$, $r^2 = 0.44$).
(PDF)

**S1 Table. Significantly changed genes along *ex vivo* CCR7⁺CD8⁺T cells.**
(PDF)

**S2 Table. Significantly changed genes along *ex vivo* CCR7-CD45RA⁺CD8⁺T cells.**
(PDF)

**S3 Table. Significantly changed genes along *ex vivo* CD28nullCD57⁺CD8⁺T cells.**
(PDF)

**S4 Table. Up-regulated genes in CCR7⁺CD8⁺ T cells compared with CCR7-CD8⁺ T cells.**
(PDF)

**S5 Table. Down-regulated genes in CCR7⁺CD8⁺ T cells compared with CCR7-CD8⁺ T cells.**
(PDF)

## Author Contributions

**Conceptualization:** Sang-Ho Lee, Byung Ha Chung.

**Funding acquisition:** Eun Jeong Ko.

**Investigation:** Jang-Hee Cho, Chan-Duck Kim, Junhee Seok, Chul Woo Yang.

**Methodology:** Eun Jeong Ko, Jung-Woo Seo, Kyoung Woon Kim, Bo-Mi Kim.

**Writing – original draft:** Eun Jeong Ko, Jung-Woo Seo, Byung Ha Chung.

**Writing – review & editing:** Sang-Ho Lee, Byung Ha Chung.

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
