## [Decision Letter · Decision Letter 0]

17 Feb 2020

PONE-D-19-34852

Phenotype and molecular signature of CD8+ T cell subsets in T cell- mediated rejections after kidney transplantation

PLOS ONE

Dear Dr. Chung,

Thank you for submitting your manuscript to PLOS ONE. After careful consideration, we feel that it has merit but does not fully meet PLOS ONE’s publication criteria as it currently stands. Therefore, we invite you to submit a revised version of the manuscript that addresses the points raised during the review process.

The manuscript is of potential interest to the renal transplant community. However, it is not acceptable for publication in its current form. There is a considerable heterogeneity of the study groups with respect to important baseline characteristics, that may have confounded the presented findings and preclude conclusions. In order to decide if this is the case I would ask you to provide more detailed both methodology and results description:

Please provide information on clinical characteristics with respect to: retansplantation, other solid organ transplantation, previous TCMR, presensitization.Please provide a comment on the significant differences between the 3 study groupsWhat was the selection process of samples ie n=32 normals + n=50 TCMR + n=39 LTGS = 121 total from the banked biosamples of the ARTKT-1 study?What were the indications for the n=91 biopsies in the normal biopsy control group in table 1?Please confirm that ARTKT-1 was used only to identify participants and access kidney tissue.  Participants were then approached for their peripheral blood. (Methods describes flow cytometry on blood within a few hours of collection).  If this is so, what is the time interval between deposition of biosamples in ARTKT-1 and the sampling of the peripheral blood?The number of participants/analyzed samples differs for various analyses - Please account for the reduced sample set, including which groups were reduced by how much and why.

There are also other issues, which are described in detail by the Reviewers.

We would appreciate receiving your revised manuscript by Apr 02 2020 11:59PM. To enhance the reproducibility of your results, we recommend that if applicable you deposit your laboratory protocols in protocols.io, where a protocol can be assigned its own identifier (DOI) such that it can be cited independently in the future. For instructions see: http://journals.plos.org/plosone/s/submission-guidelines#loc-laboratory-protocols

We look forward to receiving your revised manuscript.

Kind regards,

Justyna Gołębiewska

Academic Editor

PLOS ONE

Journal Requirements:

Please ensure that your manuscript meets PLOS ONE's style requirements, including those for file naming. The PLOS ONE style templates can be found at http://www.plosone.org/attachments/PLOSOne_formatting_sample_main_body.pdf and http://www.plosone.org/attachments/PLOSOne_formatting_sample_title_authors_affiliations.pdf

Reviewers' comments:

Reviewer's Responses to Questions

**Comments to the Author**

1. Is the manuscript technically sound, and do the data support the conclusions?

Reviewer #1: Yes

Reviewer #2: No

2. Has the statistical analysis been performed appropriately and rigorously? 

Reviewer #1: Yes

Reviewer #2: Yes

3. Have the authors made all data underlying the findings in their manuscript fully available?

Reviewer #1: Yes

Reviewer #2: Yes

4. Is the manuscript presented in an intelligible fashion and written in standard English?

Reviewer #1: Yes

Reviewer #2: Yes

5. Review Comments to the Author

Reviewer #1: Research pursues one of the universal aims of kidney transplantation: cells in the peripheral blood tell us the pathobiological processes in the allograft.

Strengths of the research

• access to samples – over 10 years clinical follow up, sound control groups, access to the ARTKT-1 study biosamples

• good representation of CD8 subsets –the CCR7+, CD28 null CD57+ and the CCR7- CD45RA+ subsets - and orientation in the Introduction as to why these sub-populations.

• good command of the transcriptomic methods

• limitation section is an honest assessment, including the comments on the methodological pathway in biomarker research

Inconsistencies in the internal congruence of the work:

• Sometimes cells are called T cells and sometimes CD8 cells eg in abstract CD28 null CD57+ T cells in one place and CD8+ T cells in another

• Convention of using the / in the string describing the CD8 cells is confusing because

o Applied inconsistently

o Reader has to work hard to decipher in the section about ratios of the different cell populations, see results starting line 274 where cell proportions are reported

• line 440 CD8 subset described as CD57+CD28nullCD8+ T cells and everywhere else the more common convention of CD28nullCD57+CD8+ T cells in used. Please edit

I have the following queries:

Q1 what was the selection process of samples ie n=32 normals + n=50 TCMR + n=39 LTGS = 121 total from the banked biosamples of the ARTKT-1 study?

Q2: are the patients of this study unique or have they been reported in other studies/publications?

Q3: please confirm which groups have kidney tissue as well as peripheral blood ie were the LTGS biopsied (table 1 indicates not)?

Q4: what were the indications for the n=91 biopsies in the normal biopsy control group in table 1?

Q5: there are 4 transplant centres and 3 Institutional Review Boards. Which Board has oversight of which 2 transplanting centres?

Q6: table 1 shows significant differences between the 3 study groups. Suggestion: brief commentary acknowledging this and does it have an impact on results in anyway ie older age group in LTGS?

Q7: please confirm that ARTKT-1 was used only to identify participants and access kidney tissue. Participants were then approached for their peripheral blood. (Methods describes flow cytometry on blood within a few hours of collection). If this is so, what is the time interval between deposition of biosamples in ARTKT-1 and the sampling of the peripheral blood?

Q8: micro-array and flow cytometric data were correlated for n=108. Please confirm these are discrete from the n=153 participants in the ARTKT-1 study reported in reference 8.

Q9: please confirm n=108 are a subset of the n=121 reported in table 1.

Q10: if so, why was the n=121 reduced to n=108? Which of the 3 groups was reduced and by how much?

Q11: similarly, expression data were matched to cell-phenotype data in n=101 samples. Please account for the reduced sample set, including which groups were reduced by how much and why

Q12: is the control group in the microarray studies (n=3 healthy volunteers, line 200) a subset of the control group in table 1?

Q13: Line 323 describes these as the same donors. Same donors as which group?

Q14: please confirm cells in the microarray studies are pooled as opposed to single cell RNA sequencing.

Q15: the text of line 279 does not align with Figures 2C-2D which it references where the data for the TCMR group have a lower mean. It appears the figures are reporting the inverse of the text. If this is so, suggest align the text and figures.

Q16: point of clarification – is table 3 CCR7+ CD8+ T cells as described in the text in line 338 or CCR7-CD8+ T cells as described in the legend of the table, line 397?

Q17: similarly, in table 4, line 341 reads as the comparator group is CCR7+ CD8+ cells vs the table legend, line 423 has CCR7-CD8+ cells as the comparator group

Q18: I found the text describing the results in tables 3 and 4 confusing – lines 335 to 341. Table 3 reports on 7 genes, of which 5 appear to move in the same direction as the comparator but the text in line 336 suggests otherwise. What am I missing here?

Q19: Similarly, in table 4, where the text suggests 25 genes up-regulated in the ex vivo experiments are down regulated in the in vivo experiments – lines 338 to 341 – but table 4 shows scores in the same direction

Q20: line 442 highlights the sections of Figure 4 are not contiguous ie panel D is after panel D and not panel C. Please edit

Q21: line 482 has significance in it twice. Please edit

Reviewer #2: This reviewer appreciated reading this manuscript and the efforts the authors undertook to conduct this study. Chung et al. give insights into there experience of CD8 T-cell subsets monitoring among kidney transplant recipients to predict the development of TCMR.

However, this manuscripts includes some major issues that highly dampen the enthusiasm.

- Heterogeneity of the study groups with respect to import baseline characteristics, that my have confounded the presented findings:

o Time of sampling posttransplantation: while the mean time posttransplantation in the normal control group was 6.6 months, the mean time in the TCMR group was 18.0 months. The expected impact of induction immunosuppression, high-dose initial maintenance immunosuppression, and ABO desensitization therefore may have contributed to the observed findings. Particularly the impact of thymoglobuline on effector memory T-cells impacts the early posttransplant period. After thymoglobuline induction effector memory T cells are expected to recover to pretransplant values by 3 to 6 months posttransplantation.

o One wounders why the mean time to TCMR was 18 months. Where these patients experiencing previous TCMRs and treatment for previous TCMR? The authors should consider including patients with first TCMR and only within the first posttransplant year to account for heterogeneity in the study groups.

o Cyclosporin vs Tacrolimus: while almost all patients in the normal control group were unter tacrolimus, 20% of the patients in the TCMR group were under cyclosporine. The authors should provide and reckon the impact CNI trough levels on their findings.

o More information need to be provided on clinical characteristics with respect to: retansplantation, other solid organ transplantation, previous TCMR, presensitization.

- The is a remarkable overlap of the presented findings between patients of the normal control group and patients of the TCMR group. Due to the cross-sectional study design (with samples obtained at active TCMR) no conclusions with respect to the predictive value of those biomarkers can be made.

At least in this reviewers point of view, the cross-sectional design, the heterogeneity of the study population and the lack of characterization of donor-specificity of those T-cell responses limit the novelty of the findings and the presented manuscript.

6. PLOS authors have the option to publish the peer review history of their article (what does this mean?). If published, this will include your full peer review and any attached files.

Reviewer #1: Yes: Helen G Healy

Reviewer #2: No

---

## [Author Response · Author response to Decision Letter 0]

7 Apr 2020

Dear Prof. Justyna Gołębiewska

Thank you very much for the evaluation of our manuscript. We are returning a revised manuscript which incorporates many of the suggestions made by reviewers. A response to the reviewer’s suggestions has been listed one by one, and an index of change has been included. We hope that the comments of the reviewers are adequately addressed in the revised manuscript.

Manuscript number: PONE-D-19-34852

Manuscript title:

Phenotype and molecular signature of CD8+ T cell subsets in T cell- mediated rejections after kidney transplantation

Index of changes 

Major changes: 

1. Addition of the information of clinical characteristics with respect to re-transplantation, and other solid transplantation, previous TCMR, pre-sensitization. 

2. Provide a comment on the significant differences between the 3 study groups

3. Explain about the selection process of samples

4. Addition of the explanation of ARTKT-1 cohort.

Minor changes

1. Correction of typos

Sincerely yours,

Byung Ha Chung M.D., Ph.D.,

Associated Professor, department of Internal Medicine

Seoul St. Mary’s Hospital, The Catholic University of Korea.

Phone: 82-2258--6066, Fax: 82-2-536-0323

E-mail: chungbh@catholic.ac.kr

Response to Editor’s comments

1. Please provide information on clinical characteristics with respect to: retansplantation, other solid organ transplantation, previous TCMR, pre-sensitization.

→ Thank you for your comments. We added the information about re-transplant, previous TCMR, pre-sensitization in table 1 in the revised manuscript. We did not include any case of other solid organ transplantation, so we added the description “we did not include patient who took any other solid organ transplantation” in the revised manuscript,

2. Please provide a comment on the significant differences between the 3 study groups

→ Thank you for your comments. We described the significant differences between the 3 study groups as a limitation of this study in the discussion session.

3. What was the selection process of samples ie n=32 normals + n=50 TCMR + n=39 LTGS = 121 total from the banked biosamples of the ARTKT-1 study?

→ Thank you for your comment. We selected these samples just according to availability of PBMCs samples from the banked biosamples of the ARTKT-1 study. So, we just tried to include all cases whose PBMC samples were available for this study.

4. What were the indications for the n=91 biopsies in the normal biopsy control group in table 1?

→ The number of normal biopsy was 32 not 91. The indications for normal control was “surveillance biopsy”. We included those cases when biopsy finding showed “unremarkable findings” without any evidence of acute rejection, BK virus nephropathy or any pathologic findings. 

5. Please confirm that ARTKT-1 was used only to identify participants and access kidney tissue. Participants were then approached for their peripheral blood. (Methods describes flow cytometry on blood within a few hours of collection). If this is so, what is the time interval between deposition of biosamples in ARTKT-1 and the sampling of the peripheral blood?

→ Thank you for your comments. We can confirm that ARTKT-1 was used only to identify particitipants and access kidney tissue. We added following sentence in the revised manuscript. “ARTKT-1 was used only to identify participants and access kidney tissue.”

There is some error in the description for flowcytometry. At first, cells were frozen immediately after collected at the time of allograft biopsy, and were transported to our center for flowcytometric analysis. So we omitted the sentences “In the samples used for the ex vivo study, we did flow cytometry analysis within a few hours after collection of peripheral blood.” in the revised manuscript. Deposition of biosamples in ARTKT-1 was done within 1 hour after the sampling of peripheral blood. We added this description as well.

6. The number of participants/analyzed samples differs for various analyses - Please account for the reduced sample set, including which groups were reduced by how much and why.

→ Thank you for your comments. The patients (n=109) used to investigate the association between phenotype and transcript is an independent group to the patients group (n=121) used in the analysis in this study. The PBMC samples of those 109 patients were used for previous study (Immune Netw. 2018;18(5):e36), hence additional samples were not available any more for this study any more. We just used those data to show the association between immune cell phenotype and transcript in peripheral blood of kidney transplant recipients. 

Response to reviewers’ comments: 

Reviewer #1 

Reviewer #1: Research pursues one of the universal aims of kidney transplantation: cells in the peripheral blood tell us the pathobiological processes in the allograft.

Strengths of the research

• access to samples - over 10 years clinical follow up, sound control groups, access to the ARTKT-1 study biosamples

• good representation of CD8 subsets -the CCR7+, CD28 null CD57+ and the CCR7- CD45RA+ subsets - and orientation in the Introduction as to why these sub-populations.

• good command of the transcriptomic methods

• limitation section is an honest assessment, including the comments on the methodological pathway in biomarker research

Inconsistencies in the internal congruence of the work:

• Sometimes cells are called T cells and sometimes CD8 cells eg in abstract CD28 null CD57+ T cells in one place and CD8+ T cells in another

→ Thank you for your comments. We used only CD8+ T cells around the whole manuscript in the revised manuscript.

• Convention of using the / in the string describing the CD8 cells is confusing because

o Applied inconsistently

o Reader has to work hard to decipher in the section about ratios of the different cell populations, see results starting line 274 where cell proportions are reported

→ Thank you for your comment We omitted “/” in the revised manuscript to avoid confusing.

• line 440 CD8 subset described as CD57+CD28nullCD8+ T cells and everywhere else the more common convention of CD28nullCD57+CD8+ T cells in used. Please edit

→ Thank you for your comments. We edited all terms of immune cells in the revised manuscript. 

I have the following queries:

Q1 what was the selection process of samples ie n=32 normals + n=50 TCMR + n=39 LTGS = 121 total from the banked biosamples of the ARTKT-1 study?

→ Thank you for your comment. We selected these samples just according to availability of PBMCs samples from the banked biosamples of the ARTKT-1 study. So, we just tried to include all cases whose PBMC samples were available for this study.

Q2: are the patients of this study unique or have they been reported in other studies/publications?

→ Thank you for your comment. The data using biosamples included in the ARTKT-1 study have been reported in other studies. (J Chromatogr B Analyt Technol Biomed Life Sci. 2019 Jun 15;1118-1119:157-163. Sci Rep. 2019 Feb 12;9(1):1854. PLoS One. 2018 Sep 18;13(9):e0204204. PLoS One. 2018 Jul 16;13(7):e0200631, PLoS One. 2017 Dec 21;12(12):e0190068.)

Q3: please confirm which groups have kidney tissue as well as peripheral blood ie were the LTGS biopsied (table 1 indicates not)?

→ We can confirm that ARTKT-1 was used only to identify particitipants and access kidney tissue. We added following sentence in the revised manuscript. 

“ARTKT-1 was used only to identify participants and access kidney tissue.”

Q4: what were the indications for the n=91 biopsies in the normal biopsy control group in table 1?

→ The number of normal biopsy was 32 not 91. The indications for normal control was “surveillance biopsy”. We included those cases when biopsy finding showed “unremarkable findings” without any evidence of acute rejection, BK virus nephropathy or any pathologic findings. 

Q5: there are 4 transplant centres and 3 Institutional Review Boards. Which Board has oversight of which 2 transplanting centres?

→ IRB for kyoung-hee neomedical center has oversight both Kyoung Hee University Hospital at Gangdong, Kyung Hee University Hospital. We corrected it accordingly in the revised manuscript. 

Q6: table 1 shows significant differences between the 3 study groups. Suggestion: brief commentary acknowledging this and does it have an impact on results in anyway ie older age group in LTGS?

→ Thank you for your comments. It is possible that the significant difference between three groups can have impact on results. So, we mentioned the significant heterogeneity is the limitation of this study in the revised manuscript. 

Q7: please confirm that ARTKT-1 was used only to identify participants and access kidney tissue. Participants were then approached for their peripheral blood. (Methods describes flow cytometry on blood within a few hours of collection). If this is so, what is the time interval between deposition of biosamples in ARTKT-1 and the sampling of the peripheral blood?

→ Thank you for your comments. We can confirm that ARTKT-1 was used only to identify particitipants and access kidney tissue. We added following sentence in the revised manuscript. “ARTKT-1 was used only to identify participants and access kidney tissue.”

There is some error in the description for flowcytometry. At first, cells were frozen immediately after collected at the time of allograft biopsy, and were transported to our center for flowcytometric analysis. So we omitted the sentences “In the samples used for the ex vivo study, we did flow cytometry analysis within a few hours after collection of peripheral blood.” in the revised manuscript. Deposition of biosamples in ARTKT-1 was done within 1 hour after the sampling of peripheral blood. We added this description as well.

Q8: micro-array and flow cytometric data were correlated for n=108. Please confirm these are discrete from the n=153 participants in the ARTKT-1 study reported in reference 18.

→ Thank you for your comments. These 108 cases are a subset of n=153 participants in the ARTKT-1 study in reference 18 (Immune Netw. 2018;18(5):e36). As we described in the manuscript, “we used microarray and flowcytometry data of 108 KTRs in whom the data for CCR7+CD8+ T, CCR7-CD45RA+CD8+ T, and CD28nullCD57+CD8+ T cells were available.”

In other words, both flowcytometric and microarray data were available in those 108 patients out of 153 participants..

Q9: please confirm n=108 are a subset of the n=121 reported in table 1.

→ The patients (n=109) used to investigate the association between phenotype and transcript is an independent group to the patients group (n=121) used in the analysis in this study. The PBMC samples of those 109 patients were used for previous study (Immune Netw. 2018;18(5):e36), hence additional samples were not available any more for this study any more. We just used those data to show the association between immune cell phenotype and transcript in peripheral blood of kidney transplant recipients. 

Q10: if so, why was the n=121 reduced to n=108? Which of the 3 groups was reduced and by how much?

→ As we mentioned above (for Q9), The patients (n=108) used to investigate the association between phenotype and transcript is independent group to the patients group (n=121) used in the analysis in this study.

Q11: similarly, expression data were matched to cell-phenotype data in n=101 samples. Please account for the reduced sample set, including which groups were reduced by how much and why

→ As we mentioned above (for Q9), The patients (n=108) used to investigate the association between phenotype and transcript is independent group to the patients group (n=121) used in the analysis in this study.

Q12: is the control group in the microarray studies (n=3 healthy volunteers, line 200) a subset of the control group in table 1? 

→No, we used blood samples from healthy volunteers. They were kidney transplant recipients. They were just healthy volunteers and so not were included in table 1

Q13: Line 323 describes these as the same donors. Same donors as which group?

→ Same donor means that we collected PBMCs for the microarray analysis on CCR7+CD8+ T cells and also CCR7-CD8+ T cells from same donor.

Q14: please confirm cells in the microarray studies are pooled as opposed to single cell RNA sequencing.

→ Yes, we pooled cells for microarray as opposed to single cell RNA sequencing. We added this description in the revised manuscript.

Q15: the text of line 279 does not align with Figures 2C-2D which it references where the data for the TCMR group have a lower mean. It appears the figures are reporting the inverse of the text. If this is so, suggest align the text and figures.

→ Thank you for your comments. We revised it accordingly as follows 

“it was significantly lower in the TCMR group than in the NC or LTGS group.”

Q16: point of clarification - is table 3 CCR7+ CD8+ T cells as described in the text in line 338 or CCR7-CD8+ T cells as described in the legend of the table, line 397?

→ Thank you for your comments. It was error. We corrected it to CCR7+CD8+ T

Q17: similarly, in table 4, line 341 reads as the comparator group is CCR7+ CD8+ cells vs the table legend, line 423 has CCR7-CD8+ cells as the comparator group

→ Thank you for your comments. It was error. We corrected it to CCR7+CD8+ T

Q18: I found the text describing the results in tables 3 and 4 confusing - lines 335 to 341. Table 3 reports on 7 genes, of which 5 appear to move in the same direction as the comparator but the text in line 336 suggests otherwise. What am I missing here?

→ Thank you for your comments. 

“Out of the eight up-regulated genes along with the ex vivo CCR7-CD45+CD8+ T cells~” 

Above eight up-regulated genes are presented in Table S2, and out of them, “five genes were included in the down-expressed genes on in vitro CCR7+CD8+ T cells” as shown in table 3. We indicated ‘Table S2’ in the revised manuscript to avoid confusion.

Q19: Similarly, in table 4, where the text suggests 25 genes up-regulated in the ex vivo experiments are down regulated in the in vivo experiments - lines 338 to 341 - but table 4 shows scores in the same direction

→ Thank you for your comments. Similarly, we indicated ‘Table S3’ in the revised manuscript as follows.

“25 out of 124 up-regulated genes and 10 out of 19 down-regulated genes along with ex vivo CD28nullCD57+CD8+ T cells (Table S3) were included in the down-regulated or up-regulated genes on in vitro CCR7+CD8+ T cells, respectively, as shown in Table 4.

Q20: line 442 highlights the sections of Figure 4 are not contiguous ie panel D is after panel D and not panel C. Please edit

→ This legend is confusing, so we revised it as follows

“Combination of the ratio between CCR7+CD8+ T vs CD28nullCD57+CD8+ T and the ratio between CCR7+CD8+ T vs CD45RA+CCR7-CD8+ T”

Q21: line 482 has significance in it twice. Please edit

→ Thank you for your comment, We edit it

Reviewer #2: This reviewer appreciated reading this manuscript and the efforts the authors undertook to conduct this study. Chung et al. give insights into there experience of CD8 T-cell subsets monitoring among kidney transplant recipients to predict the development of TCMR.

However, this manuscripts includes some major issues that highly dampen the enthusiasm.

- Heterogeneity of the study groups with respect to import baseline characteristics, that may have confounded the presented findings:

→ Thank you for your comments. We described the significant differences between the 3 study groups as a limitation of this study in the discussion session.

o Time of sampling posttransplantation: while the mean time posttransplantation in the normal control group was 6.6 months, the mean time in the TCMR group was 18.0 months. The expected impact of induction immunosuppression, high-dose initial maintenance immunosuppression, and ABO desensitization therefore may have contributed to the observed findings. Particularly the impact of thymoglobuline on effector memory T-cells impacts the early posttransplant period. After thymoglobuline induction effector memory T cells are expected to recover to pretransplant values by 3 to 6 months posttransplantation.

→ Thank you for your comments. We described above comments as limitation of this study in the revised manuscript.

o One wounders why the mean time to TCMR was 18 months. Where these patients experiencing previous TCMRs and treatment for previous TCMR? The authors should consider including patients with first TCMR and only within the first posttransplant year to account for heterogeneity in the study groups.

→ Thank you for your comments. We excluded 33 cases who suffered previous TCMR or late TCMR (> 1 post-transplant year) in TCMR group, hence we performed additional analysis only for the first TCMR cases occurred within 1 post-transplant year. But, as shown in the below figure, the results showed very similar pattern to that presented in the original manuscript. 

o Cyclosporin vs Tacrolimus: while almost all patients in the normal control group were under tacrolimus, 20% of the patients in the TCMR group were under cyclosporine. The authors should provide and reckon the impact CNI trough levels on their findings.

→ Thank you for your comments, We added above contents as limitation of this study in the revised manuscript. 

o More information need to be provided on clinical characteristics with respect to: retansplantation, other solid organ transplantation, previous TCMR, presensitization.

→ Thank you for your comments. We added the information about re-transplant, previous TCMR, pre-sensitization in table 1 in the revised manuscript. We did not include any case of other solid organ transplantation, so we added the description “we did not include patient who took any other solid organ transplantation” in the revised manuscript,

- The is a remarkable overlap of the presented findings between patients of the normal control group and patients of the TCMR group. Due to the cross-sectional study design (with samples obtained at active TCMR) no conclusions with respect to the predictive value of those biomarkers can be made.

→ Thank you for your comments and we absolutely agree with your opinion. Therefore, we described the limitation of this study as follows ; “We analyzed samples taken from a cross-sectional cohort; hence we did not investigate the dynamic changing pattern of each cell type. It will be necessary to observe the change of each cell type in a prospective cohort.”

At least in this reviewers point of view, the cross-sectional design, the heterogeneity of the study population and the lack of characterization of donor-specificity of those T-cell responses limit the novelty of the findings and the presented manuscript.

→ Thank you for your important comments. We added your comments as the limitation of this study in the revised manuscript. “Lastly, three groups showed significant heterogeneity in terms of clinical characteristics, and also we did not show donor-specificity of T cell responses, which can limit the novelty of this study.”

---

## [Decision Letter · Decision Letter 1]

26 May 2020

Phenotype and molecular signature of CD8+ T cell subsets in T cell- mediated rejections after kidney transplantation

PONE-D-19-34852R1

Dear Dr. Chung,

We are pleased to inform you that your manuscript has been judged scientifically suitable for publication and will be formally accepted for publication once it complies with all outstanding technical requirements.

With kind regards,

Justyna Gołębiewska

Academic Editor

PLOS ONE

Additional Editor Comments (optional):

Reviewers' comments:

Reviewer's Responses to Questions

**Comments to the Author**

1. If the authors have adequately addressed your comments raised in a previous round of review and you feel that this manuscript is now acceptable for publication, you may indicate that here to bypass the “Comments to the Author” section, enter your conflict of interest statement in the “Confidential to Editor” section, and submit your "Accept" recommendation.

Reviewer #1: All comments have been addressed

Reviewer #3: All comments have been addressed

2. Is the manuscript technically sound, and do the data support the conclusions?

Reviewer #1: Yes

Reviewer #3: Yes

3. Has the statistical analysis been performed appropriately and rigorously? 

Reviewer #1: Yes

Reviewer #3: Yes

4. Have the authors made all data underlying the findings in their manuscript fully available?

Reviewer #1: Yes

Reviewer #3: Yes

5. Is the manuscript presented in an intelligible fashion and written in standard English?

Reviewer #1: Yes

Reviewer #3: Yes

6. Review Comments to the Author

Reviewer #1: one question

- line 285, figure 2C as is or should it be 2E?

one edit

- line 254, delete out of at the beginning of the line

one annoyance

- legend in figure 4 CD descriptions of the cell sub-populations in B), C), E) and F) are not consistent with the descriptors in the text eg CD45RA+CCR7-CD8+ compared with CCR7-CD45RA+CD8+ in the text. This was raised in the first round of review and addressed in the text but remains in this one place.

one comment

- graphics do not reproduce well in the internet version

Reviewer #3: The authors have addressed an important issue in T cell biology. This is a revised version and the authors have responded to earlier cirques and provided appropriate chnages. I am satisfied with their changes.

7. PLOS authors have the option to publish the peer review history of their article (what does this mean?). If published, this will include your full peer review and any attached files.

Reviewer #1: Yes: Helen G Healy

Reviewer #3: Yes: Manikkam Suthanthiran

---

## [Editor Report · Acceptance letter]

4 Jun 2020

PONE-D-19-34852R1 

Phenotype and molecular signature of CD8^+^ T cell subsets in T cell- mediated rejections after kidney transplantation 

Dear Dr. Chung:

I'm pleased to inform you that your manuscript has been deemed suitable for publication in PLOS ONE. Congratulations! Your manuscript is now with our production department. 

Kind regards, 

on behalf of

Dr. Justyna Gołębiewska 

Academic Editor

PLOS ONE